# Inflammation in Obesity-Related Complications in Children: The Protective Effect of Diet and Its Potential Role as a Therapeutic Agent

**DOI:** 10.3390/biom10091324

**Published:** 2020-09-16

**Authors:** Valeria Calcaterra, Corrado Regalbuto, Debora Porri, Gloria Pelizzo, Emanuela Mazzon, Federica Vinci, Gianvincenzo Zuccotti, Valentina Fabiano, Hellas Cena

**Affiliations:** 1Pediatric and Adolescent Unit, Department of Internal Medicine, University of Pavia, 27100 Pavia, Italy; 2Pediatric Unit, “V. Buzzi” Children’s Hospital, 20153 Milan, Italy; gianvincenzo.zuccotti@unimi.it (G.Z.); valentina.fabiano@unimi.it (V.F.); 3Pediatric Unit, Fond. IRCCS Policlinico S. Matteo and University of Pavia, 27100 Pavia, Italy; corrado.regalbuto01@universitadipavia.it (C.R.); fede90vinci@gmail.com (F.V.); 4Laboratory of Dietetics and Clinical Nutrition, Department of Public Health, Experimental and Forensic Medicine, University of Pavia, 27100 Pavia, Italy; debora.porri01@universitadipavia.it (D.P.); hellas.cena@unipv.it (H.C.); 5“L. Sacco” Department of Biomedical and Clinical Science, University of Milan, 20153 Milan, Italy; gloria.pelizzo@unimi.it; 6Pediatric Surgery Unit, “V. Buzzi” Children’s Hospital, 20153 Milan, Italy; 7IRCCS Centro Neurolesi “Bonino-Pulejo”, 98124 Messina, Italy; emanuela.mazzon@irccsme.it; 8Clinical Nutrition and Dietetics Service, Unit of Internal Medicine and Endocrinology, ICS Maugeri IRCCS, 27100 Pavia, Italy

**Keywords:** obesity, children, nutrients, diet, inflammation, prevention, complications

## Abstract

Obesity is a growing health problem in both children and adults, impairing physical and mental state and impacting health care system costs in both developed and developing countries. It is well-known that individuals with excessive weight gain frequently develop obesity-related complications, which are mainly known as Non-Communicable Diseases (NCDs), including cardiovascular disease, type 2 diabetes mellitus, metabolic syndrome, non-alcoholic fatty liver disease, hypertension, hyperlipidemia and many other risk factors proven to be associated with chronic inflammation, causing disability and reduced life expectancy. This review aims to present and discuss complications related to inflammation in pediatric obesity, the critical role of nutrition and diet in obesity-comorbidity prevention and treatment, and the impact of lifestyle. Appropriate early dietary intervention for the management of pediatric overweight and obesity is recommended for overall healthy growth and prevention of comorbidities in adulthood.

## 1. Introduction

Obesity is a growing health problem in both children and adults, not only impacting people’s physical and mental health but also the economy of most societies.

According to the World Health Organization (WHO), overweight and obesity in children under 5 years of age are defined as weight-for-height 2 and 3 standard deviations (SD), respectively, above the WHO Child Growth Standards reference median; for children aged 5–19 years, overweight and obesity are defined as BMI-for-age 1 SD and 2 SD, respectively, above the WHO Growth Standards reference median [1].

In 2016, the WHO stated that nearly 41 million children below the age of 5 and over 340 million children and adolescents between 5 and 19 years of age were either overweight or affected by obesity. The WHO estimated that 25–70% and 5–30% of the European population were overweight or obese, respectively [2]. The prevalence of obesity among adults in the United States of America is around 33.8% for males and 35.5% for females, while the prevalence of overweight and obesity in children between 2 and 12 years of age is around 16%, and in adolescents is 17.6% [3].

Prevention of childhood obesity is critical to preventing complications in adolescence, early adulthood and adulthood [4]. Indeed, it is well-known that individuals with obesity are likely to develop related complications, including cardiovascular disease, type 2 diabetes mellitus (T2DM), metabolic syndrome, non-alcoholic fatty liver disease (NAFLD), hypertension, hyperlipidemia and other conditions associated with chronic inflammation, which cause disability and shorten life span [5].

This narrative review aims to describe the most up-to-date evidence on complications related to inflammation in pediatric obesity and the role of diet in the inflammatory process, in order to raise awareness of lifestyle impact and to emphasize nutrition-related interventions for preventing obesity and related complications.

## 2. Methods

Since our review is intended to be narrative, no systematic search of the literature was performed; each author identified and critically reviewed the most relevant papers in the English literature regarding pediatric obesity and related inflammatory complications and unbalanced dietary patterns. The following keywords were used to search for papers published up to May 2020 in each author’s field of expertise: childhood obesity OR pediatric obesity; obesity-related cardiovascular complications; obesity-related endocrine system complications, obesity-related asthma, pathogenesis of NAFLD/NASH, ectopic lipid accumulation; adipose tissue-associated inflammation; pro-inflammatory responses; dietary patterns; nutrients; nutrition interventions; lifestyle. The following electronic databases were searched: PubMed, Scopus, EMBASE and Web of Science. The contributions were collected, and the resulting draft was discussed among authors to provide a theoretical point of view, considered an important educational tool in continuing medical education [6]. The final version was then recirculated and all the contents approved by all the co-authors.

## 3. Adipose Tissue-Associated Inflammation

Obesity is associated with chronic low-grade systemic inflammation, as it is well-defined by alterations of circulating levels of cytokines and acute phase reactants [5,6,7]. The main difference between the systemic inflammation caused by obesity and the classic pathway of inflammation is that the most important organ involved is the adipose tissue itself. Adipokines secreted by adipose tissue are involved in an autocrine and paracrine manner in the regulation of energy expenditure, insulin sensitivity, glucose and lipid metabolism, endothelial function, and inflammation [8].

Adipose tissue contains a wide variety of immune cells, which essentially turns it into an immune organ connecting metabolism and immunity. At the beginning of the 21st century, the increased secretion of cytokines, especially TNF-α and IL-6, which contribute to insulin resistance, was proved to be linked to infiltration and accumulation of macrophages in adipose tissue [9].

The central role of the macrophage as an immune cell contributing to adipose tissue inflammation has become more relevant in recent years, along with the morphological changes in the adipocyte and the alterations in the quantity and composition of immune cells, stimulating a condition of chronic low-grade inflammation.

The inflammatory cascade, as a primary immune response, is a function carried out mainly by macrophages in synergy with the toll-like receptor family, especially toll-like receptor 4 (TLR4), which binds the ligand lipopolysaccharide (LPS), initiating the signaling pathway [10]. It generates the production of the nuclear factor kB (NFkB), along with pro-inflammatory cytokines like IL-1, IL-6, and TNF-α as well as serum amyloid A3 (SAA3), alpha l-acid glycoprotein, the lipocalin 24p3 and plasminogen activator inhibitor-1 (PAI-1). 

Macrophages are peculiar in that they adapt to changing habitats, ranging from anti-inflammatory to pro-inflammatory, while maintaining tissue homeostasis [11].

There are two main types of macrophages involved in the inflammatory response: firstly, the so-called M1- macrophages or F4/80 + CD11c +, characterized by high expression of TNF-α and inducible nitric oxide synthase (iNOS), also found in animal models, primarily expressed in the visceral adipose tissue (cc Adipose tissue macrophages: going off track during obesity), and secondly, the M2 phenotype, identified as F4/80 + CD 206 + CD301 + CD11c-- macrophages, mainly prevalent in lean adipose tissue, which express genes encoding anti-inflammatory cytokines like IL-10 [11].

Macrophages found in visceral adipose tissue are predominantly M1-macrophages, primarily involved in the inflammatory response, which induce the production of other pro-inflammatory cytokines (such as TNF-α), and surround the adipocyte in a crown-like structure [12]. On the other hand, the macrophages found in lean fat are mainly the M2 anti-inflammatory type. There is also evidence that other cells, such as mast cells and neutrophils (neutrophil elastase-enhanced activity has been noted in the serum of obese patients), participate in the activation of the innate immune response.

Adaptive immune cells in animal models were found to be involved in adipose tissue inflammation, through the accumulation of both T and B cells, as shown in obese mice [13,14]. These studies also showed that there are elevated levels of IFN-γ and chemokines like CCL5 and CXCL1, which participate in the recruitment of macrophages, thus contributing to the inflammatory state of the adipose tissue. The role of TNF-α is also central; it stimulates the production of IL-6, which is a soluble factor with several effects on inflammation, along with immune response and hematopoiesis. TNF-α leads to insulin resistance by means of phosphorylation of serine residues of the insulin receptor substrate-1, preventing the receptor substrate from binding to the insulin receptor, and inhibiting insulin action. Once IL-6 has been produced, it moves to the liver through the bloodstream, stimulating the production of acute phase reactants such as C-reactive protein (CRP). CRP is the most frequently measured acute phase protein, as it is known to be one of the main actors in the inflammatory cascade [15]. It activates and sustains the production of other inflammatory cytokines which mediate and mutually induce the production of CRP itself, maintaining and prolonging the inflammatory state along with tissue factor activation and complement activation, which are linked to the thrombogenesis pathway.

CRP is probably one of the most commonly used markers in assessing systemic inflammation. It is produced by the liver, peripheral leukocytes and the adipose tissue in response to multiple triggers, particularly IL-6 and other systemic inflammatory cytokines. CRP has specific peripherical roles, including the activation of phagocytic cells through the FcγRIIA receptor. Given that visceral adipose tissue is associated with a proinflammatory state, it is clear why CRP increases in the case of central obesity.

A second (late) cytokine response subsequently downregulates the first cascade, lowering the inflammatory state and diminishing the concentration of inflammatory cytokines and acute phase reactants.

Adipokines dysregulation has also emerged as a characteristic of chronic inflammation. Adiponectin and leptin are two hormones produced by the adipose tissue. Leptin, known to be an appetite-suppressive hormone that increases during inflammation, is also involved in the Th1 immune response, which sustains the inflammatory state [16]. On the contrary, adiponectin inhibits the NFkB signaling pathway and also protects against insulin resistance and atherosclerosis. It also antagonizes the TNF-α and its concentration decreases in obese subjects. It is secreted by adipocytes in inverse proportion to the amount of lipid stored and seems to act on insulin sensitivity in obese animal models. Low levels of adiponectin and increased insulin resistance are also known to be linked to the clinical features of metabolic syndrome. Remarkably, low adiponectin concentration is associated with high levels of inflammatory cytokines, while, in contrast, infusions of adiponectin in animal models result in reduced systemic inflammation, through mechanisms which remain unclear.

The changes in cellular nutrient homoeostasis found in obese subjects affect the function of the mitochondria and the endoplasmic reticulum [17]. Mitochondrial functions include energy pathways, cellular redox homoeostasis, calcium buffering and regulation of apoptosis; all these mechanisms are crucial to maintaining cellular bioenergetics and integrity and they may be involved in the development of inflammation [17]. In particular, the importance of calcium buffering will be explained later, in reference to cardiovascular disease. 

## 4. Pediatric Obesity and Related Inflammatory Complications

Pediatric obesity is a multisystem condition that has potentially harmful consequences and various complications, including hyperinsulinemia and insulin resistance (identified in children as young as 5 years of age), hyperandrogenism and polycystic ovarian syndrome, hypertension, dyslipidemia, chronic inflammation, increased blood clotting tendency, endothelial dysfunction, asthma, and gastrointestinal and neurological disorders [18].

Some studies have evaluated the effects pediatric obesity and its complications may have on life span, predicting a shorter life expectancy [19]. The chronic inflammatory status associated with obesity plays a crucial role in the development of complications. Recent evidence demonstrates that the initiating events in obesity-induced inflammation start early in childhood [20]. 

### 4.1. Endocrine System

Children affected by overweight or obesity are hyperinsulinemic and have approximately 40% less insulin-stimulated glucose compared with normal-weight children. The development of T2DM is closely related to the insulin resistance value [4,5,17,21]. Glucose homeostasis depends on the equilibrium between insulin secretion by the pancreatic cells and insulin activity. The evolution of impaired glucose tolerance in subjects with obesity is connected to the worsening of insulin resistance and is an intermediate stage in the natural progression to T2DM. During childhood, especially the pubertal period, there is a normal increase in insulin resistance, resulting in hyperinsulinemia, which can then normalize due to the changes at the end of puberty. Subjects with T2DM show impaired insulin activity along with insulin secretory defect. As described in detail previously, the enlarging adipose tissue in obese subjects synthesizes and secretes hormones and proteins like leptin, adiponectin and TNF-α, which modify insulin secretion and sensitivity, causing insulin resistance [9].

TNF-α and IL-6 can alter insulin sensitivity and stimulate the phosphorylation of serine residues instead of tyrosine in insulin receptor substrate-1 (IRS-1), thus inhibiting the activation of insulin signaling and sustaining insulin resistance. Therefore, it appears that such conditions lead to a vicious cycle, as hyperglycemia also induces IL-6 (proinflammatory cytokine) production from endothelium and macrophages. Furthermore, it intensifies the activity of the suppressors of cytokine signaling (SOCS) proteins, impairing insulin release and intracellular signaling pathways [21].

Obesity also plays an important role in the pathogenesis of hyperandrogenism and polycystic ovarian syndrome (PCOS). One of the underlying mechanisms might be insulin resistance, whereby the consequent hyperinsulinemia activates excess ovarian androgen production, leading to PCOS [22]. Given that obesity and weight gain are strongly associated with insulin resistance and that obesity/overweight is often present in PCOS, becoming overweight, especially during the pre-pubertal period, could be a significant factor in the development of PCOS, especially among females who have a genetic predisposition to PCOS.

Insulin, together with luteinizing hormone secretion, enhances ovarian androgen production. Abdominal fat distribution also significantly influences androgen and estrogen metabolism and the increased levels of androgens associated with abdominal fat have been correlated with menstrual irregularities and hirsutism, both of which are expressions of PCOS [23].

Additionally, several studies have demonstrated that pro-inflammatory cytokines and reactive oxygen species alter estrous cyclicity, steroidogenesis and ovulation [24]. Inflammation and oxidative stress also impair meiotic and cytoplasmic oocyte maturation. As reported by Skaznik-Wikiel et al. [25], the abnormalities in ovarian function have been correlated with increased infiltration of the ovary by macrophages, increased expression and signaling of pro-inflammatory cytokines [26,27] and increased incidence of ovarian fibrosis. An increased expression of TNF-α, IL-6 and IL-8 and activity of their associated inflammatory signaling pathways have also been found in the ovaries of obese women and animal models [26,27,28]. Finally, the role of obesity-dependent oxidative stress in the ovary [29] should also be considered.

### 4.2. Cardiovascular System

Excess-weight-related complications like dyslipidemia, hypertension and insulin resistance lead to cardiovascular complications as well.

Studies have shown that the atherosclerotic mechanism begins in childhood [30]. As regards children, atherosclerosis is generally sub-clinical, and progression is usually slow. However, fatty streaks and fibrous plaques found in children who died from other causes have been associated with high BMIs [31,32,33].

As with adults, children with obesity have raised blood pressure values. Rosner et al. [34] showed that the odds ratio of hypertension associated with overweight ranges from 2.5 to 3.7 in children and adolescents. The long-term impact in pediatric subjects is not entirely clear, although being overweight during childhood is known to be related to high levels of insulin, lipids and blood pressure in young adulthood.

The presence of atherosclerotic lesions in both the aorta and coronary arteries has been found to closely relate to obesity and overweight in pediatric subjects. Moreover, being overweight during childhood is a strong predictor of coronary calcium in adulthood [30,31,35]. The latter is a marker for plaque formation in the coronary arteries and correlates with increased risk of myocardial infarction.

At present, cardiovascular disease is perceived as a chronic inflammatory state of the vessel wall that results from the transcytosis of atherogenic apolipoproteins-B VLDL (Very Low Density Lipoprotein), IDL (Intermediate Density Lipoprotein) and LDL (Low-density lipoproteins) from the plasma to the inner layer of the vessel (intima). These lipoproteins are retained in the subendothelial space, leading to the infiltration of macrophages and to T cells interacting with each other and with the cells of the arterial wall. It is conceivable that inflammation induced by obesity accelerates the atherosclerosis pathway [36]. 

As regards HDL cholesterol, apolipoprotein A1, one of the most important constituents of the HDL molecule, interacts with the ATP-binding protein-1 promoting cholesterol efflux from macrophages to the liver, which is one of the main functions of HDL [30,31]. Despite this prevalent function, recent studies have highlighted the complex role of the HDL particle in the inflammatory process. HDL prevents the release of TNF-α and IL-1β from activated monocytes. During the acute-phase cascade, HDL molecules act as proinflammatory cells and promote the activation and migration of monocytes induced by LDL [30,31].

Furthermore, oxidative modification of apolipoprotein A1 lowers the efficiency of reverse cholesterol transport.

The conclusion of all these studies is that HDL cells probably play a dual role in atherosclerosis, in both preventing and promoting harmful athero-inflammatory responses. Thus, it is evident that a balance between the two usually exists and that obesity probably leads to a shift towards an unbalanced pathway, promoting atherosclerosis.

With regard to acute phase reactants, high levels of CRP in obese subjects seem to be not only a risk marker but also indicative of an active role in cardiovascular disease. This has been demonstrated using animal models of atherosclerosis in which human CRP is increased experimentally, resulting in worsening of arterial thrombosis and endothelial injury repair. High levels of CRP are also associated with myocardial infarction, thus emphasizing the pervasive relationship between inflammation and heart disease [37].

It has been underlined that some of the most important CVD risk factors like hypertension and hyperglycemia promote harmful stimulation for the production of leukocyte adhesion molecules, which then triggers an inflammatory response involved in the mechanism of atherosclerotic plaque formation and expansion.

Monocytes recruited within the arterial intima differentiate into macrophages, bury low-density oxidized lipoproteins, and become laden with lipids. Continuous systemic inflammation contributes to this process with neutrophils infiltrating the plaque and releasing reactive oxygen species (ROS) and eventually contributing to the growing atheroma.

Even though atherosclerosis rarely evolves to a critical point before adulthood, the process surely begins during childhood along with the increase in fatty streaks and thickening of the arterial media. An unusually high prevalence of fibrous plaque lesions in childhood has been recognized at autopsy examination, in the coronary arteries of 33% of 16 to 20-year-old adolescents who died of accidental causes [38].

### 4.3. Pulmonary System

Obesity is both a major risk factor and a disease modifier of asthma in children. Asthma and obesity in children are both characterized by chronic tissue inflammation [39].

Asthma can occasionally predispose children to obesity, on the other hand obesity can confound the diagnosis of asthma, or both can simply be present at the same time. However, most of the observational and experimental studies point to an ‘‘obese asthma’’ phenotype in which obesity modifies asthma [40,41,42,43,44].

It has been proved that children and adolescents with obesity tend to have increased asthma severity, and a decreased response to asthma medications [45,46,47].

As regards the immune pathway, TH-2 response inflammation is known to be complicated by obesity. The obese state also alters CD4 cells toward TH1 polarization, which is associated with worse asthma severity and control, along with abnormal lung function. TH17 pathways and innate lymphoid cells (ILCs) have also been implicated. Finally, in relation to eosinophilic airway inflammation, sputum eosinophil count and exhaled Nitric Oxide (NO) levels might underestimate the degree of type 2 inflammation in patients with obesity. Changes in their function in obese adipose tissue could contribute to both obesity and asthma [48,49].

Additionally, bronchoalveolar fluid levels of surfactant protein A, which helps modulate the response to infections, are lower in obese asthmatic subjects in comparison with lean patients, and mice models have shown that administration of surfactant protein A lowers lung tissue eosinophilia after allergen challenge, suggesting that changes in surfactant protein function could modify airway eosinophilia in obese asthmatic subjects.

Changes in the gut microbiome might also affect the generation of allergic airway disease [50]. Microorganism colonization of the gut is known to be important in relation to the formation of short-chain fatty acids (SCFAs). The dietary intake of patients with obesity is typically high in fat and low in soluble fiber; a low-fiber diet is associated with changes in the gut microbiome and circulating SCFA levels. Bacteroidetes bacteria, a major producer of SCFAs, are reduced in the gut of obese subjects and asthmatic patients [51].

Adipokines produced by adipose tissue can also affect the airways. Some studies have shown that increased leptin levels in obese adolescents correlate inversely with FEV1, FVC, and the FEV1/FVC ratio and that visceral fat leptin expression correlates with airway reactivity in adults. Leptin, along with adiponectin, has also been linked to exercise-induced changes in lung function [52,53,54,55,56,57].

### 4.4. Hepatobiliary System

Obesity-associated hepatobiliary diseases include non-alcoholic fatty liver disease (NAFLD) and gallbladder diseases.

NAFLD is becoming a major health problem in children with an ever-increasing incidence due to sedentary lifestyles and hyper-caloric diets, which are the main factors of the obesity epidemics [58].

Insulin signaling is a crucial factor that links intrahepatic and extrahepatic fatty acid metabolism; hepatic insulin signaling regulates pathways linked to fatty acid uptake, synthesis and storage [58,59].

Hyperinsulinemia leads to the expression of SREBP-1c, activating Lipogenic gene expression in the liver, along with the activation of ChREBP. Both of these determine the conversion of glucose into fatty acids through the enzymatic pathways [59].

One of the consequences of this increase in the synthesis of FFAs is the production of Malonil-CoA, which inhibits carnitine palmitoyltransferase-1 (CPT-1), the protein that transports FFAs within mitochondria. Concurrently, hyperinsulinemia lowers the synthesis of apolipoprotein B-100, which mobilizes triglycerides for assembly into VLDL, diminishing the triglyceride export.

These modifications determine various histological patterns, ranging from simple steatosis to Non-Alcoholic Steatohepatitis (NASH), defined by necrotic inflammation and fibrogenesis caused by an increase in the production of ROS, which cause lipid peroxidation (oxidative degradation of lipids) and consequent production of pro-inflammatory cytokines (TNF-α, TGF-β, IL-8); the latter collect neutrophils, causing necrosis and collagen synthesis [60]. This situation could worsen, evolving to cirrhosis or hepatocellular carcinoma.

As far as gallbladder diseases are concerned, most pediatric studies focus on cholelithiasis and biliary dyskinesia rather than inflammatory gallbladder disease [61]. Cholecystitis is an inflammatory disease of the gallbladder, often related to cholelithiasis, the main risk factor of which is most likely obesity [61]. Obesity leads to the formation of pro-inflammatory cytokines, known to establish a state of chronic inflammation that ultimately could lead to inflammation of the gallbladder, which in turn may result in the formation of gallstones.

### 4.5. Neurological System

Pseudotumor cerebri, also known as idiopathic intracranial hypertension (IIH), is defined by increased intracranial pressure without an intracranial pathology, and normal cerebrospinal fluid in the absence of an identifiable underlying systemic cause. Current diagnosis of IIH is based on the hallmark physical finding of papilledema on ophthalmological examination with confirmation by means of elevated cerebrospinal fluid opening pressure on lumbar puncture [62]. IIH most frequently occurs in obese women of childbearing age. In young children, IIH is equally distributed between males and females [63,64]; obesity becomes a risk factor beyond age 12, which may reflect the effect of pubertal status on the pathophysiology of primary IIH [63,64].

The mechanisms by which obesity predisposes children to IIH have not been fully elucidated, however, the link between obesity and this condition is most likely to be in the different substances, from pro-inflammatory cytokines to leptin, secreted by the adipose tissue [65].

Some studies have evaluated the correlation between the presence of higher levels of leptin in the CSF of obese patients as opposed to that of lean subjects, but its role in intracranial pressure regulation is not yet known [66].

Serum leptin are known to increase in tandem with rising BMI [67,68]. It has also been proved that the CSF serum leptin ratio decreases as BMI increases [69] and that central infusion of leptin into obese leptin-deficient mice corrects obesity [70]. In addition, leptin resistance may represent impaired hypothalamic signaling [71]. 

Thus, serum leptin concentration has been found to be significantly higher in subjects with IIH compared to both lean and obese controls [72].

Although discussing the question of cortisol in obese patients is uncommon, dysfunctions of 11β-hydroxysteroid dehydrogenase type 1(11β-HSD1) are thought to be pathological. 11β-HSD, located within the lumen of the endoplasmic reticulum, is expressed in various tissues, but its principal role is to mediate availability of local cortisol, which is also a powerful anti-inflammatory mediator, regulating adipocyte differentiation [72,73]. Abnormal hepatic 11β-HSD1 activity has been demonstrated in a number of studies (administration of cortisone acetate following an overnight dexamethasone suppression test) [74,75] and it has also been documented in subcutaneous fat taken from obese patients [76,77]. These studies suggest that 11β-HSD1 could be dysregulated in obesity and could play a role in contributing to IIH.

It is important to highlight that 11β-HSD1 activity is regulated by inflammatory cytokines (TNFα, IL-1β, IL-6) and adipokines (leptin) and an inflammatory phenotype in IIH has been proposed [72,73,78]. Hence, obesity-induced overproduction of inflammatory cytokines and adipokines may drive 11β-HSD1 activity along with abnormal glucocorticoid metabolism in IIH.

The connection between obesity and chronic low-grade inflammation, leptin/adipokines and 11β-HSD1 activity has been discussed. It is plausible that IIH, which is characterized by obesity, abnormal inflammatory response and obesity-related cytokine production may drive 11β-HSD1 activity. Consequently, increased 11β-HSD1 concentration in subjects with IIH may enhance cortisol levels at the choroid plexus and arachnoid granulation leading to altered CSF production and drainage, thus contributing to the elevated intracranial pressure [79].

### 4.6. Musculoskeletal System

Obesity is characterized by elevated lipid storage not only in subcutaneous and visceral depots, but also in non-adipose organs, a phenomenon called ectopic lipid accumulation [80]. In addition, obese subjects, whose levels of circulating FFAs are elevated, have high ectopic lipid deposition in skeletal muscles, partially resulting from increased fatty acid uptake from circulation [81,82,83,84]. 

The pool of lipids within skeletal muscle is composed of extramyocellular lipids (EMCL), localized in adipose cells between muscle fibers, and intramyocellular lipids (IMCL), located within muscle cells. IMCL are made up of triacylglycerols (TAG) and cholesterol esters, as well as lipid metabolites, like long-chain acyl CoAs, diacylglycerols and ceramides. Elevated TAG content and increased numbers of lipid droplets have been observed in muscle biopsies from obese people [85,86,87,88,89,90].

Insulin resistance, as well as mitochondrial and metabolic dysfunction, are some of the most noticeable muscle abnormalities to negatively impact whole-body metabolism and physical performance in obesity and T2DM. Skeletal muscle regeneration after injury requires the activity of muscle stem cells and satellite cells, which remain associated with skeletal muscle fibers after development.

In normal animal models, injuries cause local muscle fiber inflammation and consequent necrosis, followed by satellite cell activation, proliferation, differentiation, fusion and regrowth of muscle fibers to almost the same size as before within approximately three weeks [91].

Various studies have used myotoxins and freeze injury to assess muscle regeneration in obese or diabetic mice. In mice that were fed a high-fat diet for 8 months, lowered muscle mass after cardiotoxin injury was observed, along with smaller muscle fibers, larger interstitial spaces and increased collagen deposition, compared with lean mice [92]. Three weeks of high-fat diet in young mice resulted in a reduced concentration of satellite cells and impaired muscle regeneration after cold-induced injury [93]. Even though proliferation rates were not directly evaluated in these studies, the evidence suggests that a high concentration of adipose tissue lowers the proliferative capacity of satellite cells, either due to intrinsic metabolic properties of the muscle or alterations in circulating metabolites after high-fat feeding. 

In skeletal muscle, inflammation starts right after injury and coordinates with myogenic differentiation in order to obtain efficient muscle regeneration and regrowth [94,95]. Directly after muscle damage, there is an acute inflammatory response characterized by the infiltration of pro-inflammatory M1 macrophages which remove tissue debris. Subsequently, M2 macrophages are recruited within the inflammation site in order to reduce the inflammation through the secretion of TGF-β and IL-10, which promote myogenic differentiation [96].

In animal models, the deletion of chemokine receptor-2 (CCR-2) all but inhibits macrophage infiltration after muscle damage, resulting in inefficient muscle regeneration and regrowth [97].

In addition to that, TNF-α and cytokine IL-1 are also involved in preventing myogenic differentiation [98].

## 5. Diet and Lifestyle

Poor nutrition can profoundly affect children’s physical health, as well as their emotional and social life, contributing to atherosclerosis, obesity, metabolic syndrome, diabetes, and psychological distress [99]. Dietary attitudes and lifestyle choices, including fast eating [100] and irregular feeding with multiple meals [101], have been shown to exert potentially harmful influences on health, increasing, among other things, the onset of eating disorders [102,103].

Family habits play a key role in addressing eating behavior and physical activity [104].

High intake of processed foods and low intake of fruit and vegetables are, in fact, key factors for the development of childhood overweight and obesity, besides being a trigger for inflammation [105,106,107,108], Figure 1. Furthermore, evidence shows that Western dietary patterns can alter the composition of gut microbiota, affecting intestinal permeability and immunity, and promoting pro-inflammatory responses [109]. Processed food, rich in fats, sugars and additives and poor in fiber, may induce an adverse impact on health by modifying gut microbiota and contributing to an overgrowth of opportunistic microorganisms or pathogen species [109].

Another dietary component that affects gut microbiota is salt: processed foods contain a lot of salt and Muller DN and colleagues showed that salt pushes macrophages towards a pro-inflammatory phenotype, characterized by increased differentiation of naive CD4+ T cells into T helper (TH)-17 cells while decreasing T regulatory cell expression and anti-inflammatory activity [110]. 

The increase in the prevalence of hypertension among the young has been well documented and dietary sodium has been implicated in the association between excess adiposity and elevated blood pressure in children and adolescents [111].

Habitual Western dietary patterns tend to prefer high-energy density foods rather than nutrient-dense foods, with consequent inadequate intake of antioxidants, vitamins, minerals, fiber and ω-3 fatty acids useful in counteracting the low-grade inflammation that is typical of excessive weight gain, sedentarism and insulin resistance adiposity [112,113].

It is also worth considering that obesity can induce oxidative stress in adipocytes via production of ROS by mitochondria, which may be elevated in response to high-fat diets such as the Western diet [114].

Typically, the Western dietary pattern leads to micronutrient deficiencies (MNDs), often found in children affected by overweight or obesity as a consequence of an unbalanced and unhealthy diet [115].

Among the most significant MNDs are inadequate zinc intake and consequent deficiency. Zinc is one of hundreds of enzyme complexes involved in the metabolism of proteins, lipids, carbohydrates and nucleic acids and has well-known antioxidant properties [116].

A triple-masked, randomized, placebo-controlled trial conducted by Kelishadi and colleagues [117] found a significant decrease in Apo B/ApoA-I ratio, ox-LDL, leptin and malondialdehyde, total and LDL-cholesterol and hs-CRP in children with obesity, after they had received zinc sulfate supplementation. Zinc has demonstrated possible anti-inflammatory effects through cytokine signaling pathways and the reduction in plasma levels of IL-6, TNF-α and CRP [118,119], and this anti-inflammatory property of Zinc was recently confirmed in a randomized, double-blind clinical trial [119].

In terms of micronutrient deficiency, obesity is also characterized by low vitamin D levels, possibly due to its sequestration in adipose tissue [120]. Prevalence of vitamin D insufficiency among children and adolescents with obesity is extremely high [121]. It is well recognized that Vitamin D exhibits profound immunomodulatory functions [122,123,124] and Reyman et al. [125] described an association between vitamin D deficiency and high levels of circulating inflammatory mediators in children affected by obesity.

A recent meta-analyses addressing the association between vitamin D supplementation and systemic inflammation in adults showed that dietary cholecalciferol supplementation helps to achieve a significant reduction in the activity of inflammation, decreasing C-reactive protein and TNF-alpha and leptin concentrations [126]. However, the effects of vitamin D supplementation in children and adolescents are still poorly understood.

In recent years, omega-3 polyunsaturated fatty acids (n-3 PUFAs) have received a lot of attention.

Particularly, evidence shows that dietary fat intake is an early determinant of childhood obesity and that high dietary intake of omega-6 fatty acids leads to an increase in white adipose tissue depot and chronic inflammation [126]. On the other hand, evidence has shown that n-3 PUFAs and their derivatives, eicosapentaenoic acid (EPA) and docosahexaenoic acid (DHA), reduce plasma triglyceride levels [127] and biomarkers of inflammation including CRP [128,129,130]. Despite the fact that supplementation with omega-3 for treatment of childhood obesity has proved promising, the results are inconsistent and further studies are required to clarify timing, doses and mechanisms of action [131]. Furthermore, other nutrients and phytochemicals that are well-known for their anti-inflammatory role, including vitamins C and E, epigallocatechin gallate, lycopene, and polyphenols, have been found to modulate components of NF-κB, mitogen-activated protein kinase and IL-1β signaling [132].

Dietary guidelines designed to reduce the risk of NCDs are rich in plant-based foods, including fresh fruits and vegetables, whole grains, legumes, seeds, and nuts, and are poor in animal-source foods, particularly fatty and processed meat [133], which are associated with increased serum adiponectin concentration and decreased leptin and inflammatory markers concentrations (C-reactive protein, interleukin 6) [134,135]. Plant-based healthy dietary patterns naturally occur in certain regions of the world [136]. This is the case of the Mediterranean diet, traditionally rich in fiber from cereals such as whole-grain bread, pasta, couscous and other unrefined grains, and is rich in micronutrients and phytochemicals from fruit and vegetables of different colors and textures with many different potential health benefits [136,137,138]. Bioactive compounds like phenolic compounds, flavonoids, plant sterols, and carotenoids of plant origin are protective against chronic diseases, mainly thanks to their action upon lipid profile, endothelial function, and inflammatory mediators [139,140]. In particular, in vitro studies have amply demonstrated the beneficial effects on inflammation associated with the antioxidant activity of polyphenols, especially flavonoids such as flavonols, flavones, isoflavones, anthocyanidins, resveratrol, curcumin, tannins and lignans. In the cross-sectional HELENA study [141], authors also found an inverse association between higher intakes of total polyphenols and flavonoids and BMI [142].

Evidence suggests that dietary anti-oxidants may influence inflammatory markers linked to low-grade systemic inflammation [143] in children affected by obesity, and that specific appropriately formulated nutraceutical foods could be useful both for primary and secondary prevention.

Since greater adherence to healthy eating patterns is consistently associated with a lower risk of NCDs, interventions based on lifestyle changes, on the one hand promoting healthy dietary patterns adapted to individual food traditions and preferences, and on the other hand, increased physical activity, aimed at keeping inflammatory markers at bay, are fundamental for the successful prevention of complications due to excessive weight gain, as well as for successful medical nutrition treatment [144,145,146,147,148].

## 6. Conclusions

Obesity is associated with systemic low-grade inflammation in both children and adults, which has been acknowledged as one of the major drivers of chronic degenerative diseases [5,7].

Research shows that unbalanced dietary patterns such as the Western diet, high in simple sugars, refined carbohydrates, saturated and trans-fatty acids, lead to chronic inflammatory responses, increased adipose depot, and thus future comorbidities frequently associated with overweight and obesity [108]. Furthermore, some nutrients have distinctive effects on inflammatory response and metabolic impairment or restoration, which are likely to be mediated by nutrients that can either help release inflammatory messengers or fight against oxidative stress.

The detrimental effects of obesity on health, and the costs on health care systems, clearly dictate the need to provide nutritional interventions for preventing and treating obesity in childhood, which are known to yield positive results.

Therefore, appropriate early dietary interventions for the management of pediatric overweight and obesity are challenging but necessary, and it is advisable to start as early as possible for overall healthy growth, and prevention of comorbidities in adulthood.

## Figures and Tables

**Figure 1 biomolecules-10-01324-f001:**
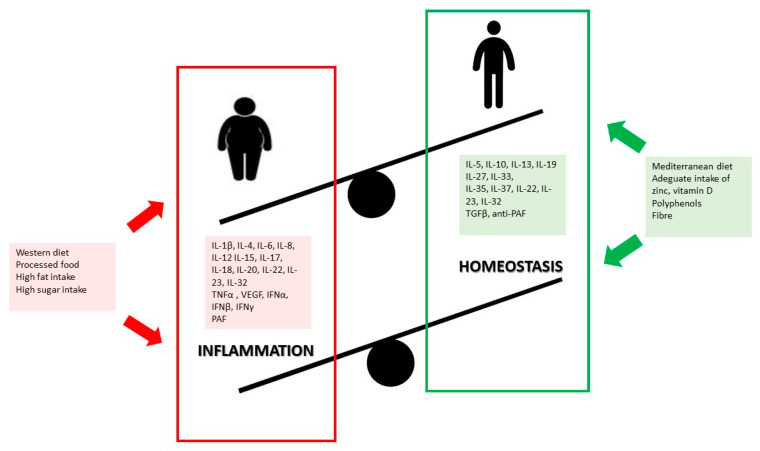
Pro-inflammatory and anti-inflammatory effects of diet in children.

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
