# Peer review of "Inflammation in Obesity-Related Complications in Children: The Protective Effect of Diet and Its Potential Role as a Therapeutic Agent"

_biomolecules, 2020, doi:10.3390/biom10091324_

Round 1

Reviewer 1 Report

Calcaterra and collaborators present a review paper surveying the problem of childhood obesity, the subjacent inflammatory causes and the multiple associated morbidities, or predisposition to later morbidities.

While not being a first review on the topic, the manuscript is informative and up-to-date, and the integration of the three main topics stated above is interesting. On the other side, the authors overindulge in citing previous reviews and extrapolating data from adult obesity to childhood obesity. Overall, it is my opinion that most of the cited review could be removed to favor a more targeted search of original work. Similarly, cited literature of studies on adults should be removed.

The authors should strive to provide a concise account on childhood obesity.

The special issue targeted by this manuscript (“Therapeutic or Preventive Potential of Dietary Compounds in the Inflammatory Response") is perhaps not fully pertinent .

Other Comments:

  • Lines 45-52: in the opening statement, the authors introduce the notion of obesity and overweight. The correspondent BMI values should be reported, and the BMI-z scores for children.
  • Throughout the manuscript, use Greek letters as appropriate for IFN and TFN
  • The paragraph “Adipose tissue-associated inflammation “ should include the original key references. As presented, it is almost exclusively citing previous review
  • Table 1 (which actually looks more like a figure), can be remove as it does not add any further information to what is presented in the main text.
  • Lines 163-170: as the authors state in line 163 (“as previously described”), the paragraph is redundant and could be removed.
  • Within the manuscript, there are often occurrences of double spaces between words. This should be amended.
  • The manuscript needs extensive editing to improve the English style. A few examples:

Line 145, replace “which” should be replaced by “that”

Line 146: “evidence” instead of “evidences” (there are several occurrences in the manuscript

Line 147: “demonstrates” instead of “demonstrating”

Line 167: “addition”, not “addiction”

Line 169: “increased” not “increasing”

Line 305: malonyl-CoA

Author Response

We thank the reviewers for their comments and suggestions. The present version of the manuscript has been revised according to their comments and criticisms.

Calcaterra and collaborators present a review paper surveying the problem of childhood obesity, the subjacent inflammatory causes and the multiple associated morbidities, or predisposition to later morbidities.

While not being a first review on the topic, the manuscript is informative and up-to-date, and the integration of the three main topics stated above is interesting. On the other side, the authors overindulge in citing previous reviews and extrapolating data from adult obesity to childhood obesity. Overall, it is my opinion that most of the cited review could be removed to favor a more targeted search of original work. Similarly, cited literature of studies on adults should be removed.

The authors should strive to provide a concise account on childhood obesity.

R: The authors thank reviewer 2 for the comments. The text has been revised and focused more on pediatric age, as suggested (red line paragraphs, pages 3-14).

The special issue targeted by this manuscript (“Therapeutic or Preventive Potential of Dietary Compounds in the Inflammatory Response") is perhaps not fully pertinent .

R: Authors are perfectly willing to change title, however before submission, the associated editor supported the pertinence of the title and article; thus we submitted the paper.

Other Comments:

  • Lines 45-52: in the opening statement, the authors introduce the notion of obesity and overweight. The correspondent BMI values should be reported, and the BMI-z scores for children.

R: Authors thank the reviewer and added the information (page 3. Lines 46-50)

  • Throughout the manuscript, use Greek letters as appropriate for IFN and TFN

R: thank you for the comment, authors corrected with Greek letters throughout the whole manuscript

  • The paragraph “Adipose tissue-associated inflammation “ should include the original key references. As presented, it is almost exclusively citing previous review

R: Authors revised the text and included original references (paragraph Adipose tissue associated inflammation

  • Table 1 (which actually looks more like a figure), can be remove as it does not add any further information to what is presented in the main text.

R: Authors have agreed to this suggestion of removing Table 1

  • Lines 163-170: as the authors state in line 163 (“as previously described”), the paragraph is redundant and could be removed.

R: Authors agree with the reviewer and removed this paragraph

  • Within the manuscript, there are often occurrences of double spaces between words. This should be amended.

R: Authors thank the reviewer for the meticulous revision and amended accordingly.

  • The manuscript needs extensive editing to improve the English style. A few examples:

Line 145, replace “which” should be replaced by “that”

Line 146: “evidence” instead of “evidences” (there are several occurrences in the manuscriptLine 167: “addition”, not “addiction”

Line 169: “increased” not “increasing”

Line 305: malonyl-CoA

R: Authors have revised English language with a native speaker (Dr. Sheila McVeigh) of the whole manuscript as recommended. We added in the Acknowledgement.

Reviewer 2 Report

This is very important to know for several populations.

However, this review has several ploblems.

1) What is this review's pourpose, could you menthion more exactly?

2) Please describe in detail how to extract the cited references.

3) For more important literature on the results, please list them specifically in the table.

4) Although it is a discussion based on the results, I think that there are many abstract expressions. Please describe more specifically.

Author Response

This is very important to know for several populations.

However, this review has several problems.

  • What is this review's pourpose, could you menthion more exactly?

R: We revised the text to clarify the purpose, as suggested by reviewer 1 (page 3, lines 63-66)

  • Please describe in detail how to extract the cited references.

R: Thanks for the suggestion, we added this information to the methods section (page 3, lines 69-82)

  • For more important literature on the results, please list them specifically in the table.

R: As narrative review, authors did not present the same data in both text and table to avoid redundancy and a waste of space and energy. However if reviewer still thinks is crucial, authors are willing to address this issue.

  • Although it is a discussion based on the results, I think that there are many abstractPlease describe more specifically.

R: It is not clear what reviewer 1 means for “abstract expressions”: does it mean that there are “theoretical expressions” in the text, or maybe that discussion includes short sentences as those usually used in abstracts? Anyway we tried to interpret this idiom and revised the discussion in light of what was already known about our subject of investigation, and to explain our new understanding of the problem after taking relevant published literature into consideration (red line paragraphs, pages 3-14).

Round 2

Reviewer 1 Report

Not applicable

Reviewer 2 Report

This revise versition of the manuscript is revised based on the reviewers comments.

So, there is no comments for this manuscript.